# Carnosic Acid Attenuates the Free Fatty Acid-Induced Insulin Resistance in Muscle Cells and Adipocytes

**DOI:** 10.3390/cells11010167

**Published:** 2022-01-05

**Authors:** Danja J. Den Hartogh, Filip Vlavcheski, Adria Giacca, Rebecca E. K. MacPherson, Evangelia Tsiani

**Affiliations:** 1Department of Health Sciences, Brock University, St. Catharines, ON L2S 3A1, Canada; dd11qv@brocku.ca (D.J.D.H.); fvlavcheski@brocku.ca (F.V.); rmacpherson@brocku.ca (R.E.K.M.); 2Centre for Bone and Muscle Health, Brock University, St. Catharines, ON L2S 3A1, Canada; 3Department of Physiology, University of Toronto, Toronto, ON M5S 1A8, Canada; adria.giacca@utoronto.ca; 4Department of Medicine, University of Toronto, Toronto, ON M5S 1A8, Canada; 5Institute of Medical Sciences, University of Toronto, Toronto, ON M5S 1A8, Canada; 6Banting and Best Diabetes Centre, University of Toronto, Toronto, ON M5S 1A8, Canada; 7Centre for Neuroscience, Brock University, St. Catharines, ON L2S 3A1, Canada

**Keywords:** muscle cells, 3T3 L1 adipocytes, insulin resistance, free fatty acid, carnosic acid, IRS-1, JNK, mTOR, p70S6K, GLUT4, AMPK

## Abstract

Elevated blood free fatty acids (FFAs), as seen in obesity, impair insulin action leading to insulin resistance and Type 2 diabetes mellitus. Several serine/threonine kinases including JNK, mTOR, and p70 S6K cause serine phosphorylation of the insulin receptor substrate (IRS) and have been implicated in insulin resistance. Activation of AMP-activated protein kinase (AMPK) increases glucose uptake, and in recent years, AMPK has been viewed as an important target to counteract insulin resistance. We reported previously that carnosic acid (CA) found in rosemary extract (RE) and RE increased glucose uptake and activated AMPK in muscle cells. In the present study, we examined the effects of CA on palmitate-induced insulin-resistant L6 myotubes and 3T3L1 adipocytes. Exposure of cells to palmitate reduced the insulin-stimulated glucose uptake, GLUT4 transporter levels on the plasma membrane, and Akt activation. Importantly, CA attenuated the deleterious effect of palmitate and restored the insulin-stimulated glucose uptake, the activation of Akt, and GLUT4 levels. Additionally, CA markedly attenuated the palmitate-induced phosphorylation/activation of JNK, mTOR, and p70S6K and activated AMPK. Our data indicate that CA has the potential to counteract the palmitate-induced muscle and fat cell insulin resistance.

## 1. Introduction

Insulin is a key player in maintaining blood glucose homeostasis. An increase in postprandial blood glucose levels results in insulin release by the pancreatic β cells. Through circulation insulin quickly reaches its target tissues, namely skeletal muscle, adipose, and liver tissue. Insulin inhibits endogenous glucose production in the liver while in skeletal muscle and adipose tissue, promotes the transport, utilization, and storage of glucose [1,2]. The result of insulin’s actions is to maintain the plasma glucose levels within a physiological range of 4–7 millimolar (mM).

Insulin acts on its target tissues, including muscle and fat cells, by binding first to its receptor. This initiates a series of cascades including tyrosine phosphorylation of the receptor and insulin receptor substrate (IRS-1), and downstream activation of the phosphatidylinositol-3 kinase (PI3K)/protein kinase B/Akt signaling axis resulting in glucose transporter (GLUT4) translocation from an intracellular pool to the plasma membrane and increased glucose uptake [1,2]. Impairments in the IRS-1-PI3K-Akt cascade contribute to insulin resistance and type 2 diabetes mellitus (T2DM) [1,2,3,4,5].

Insulin resistance and T2DM are strongly correlated with increased plasma lipid levels and obesity. In vitro studies have shown that exposure of muscle and fat cells/adipocytes to free fatty acids (FFA) such as palmitate, results in insulin resistance [6,7]. Additionally, evidence from in vivo animal studies has shown that lipid infusion [8,9], or increased plasma lipid levels by high-fat feeding, results in muscle and fat insulin resistance [8,10,11,12]. Evidence indicates that serine phosphorylation of IRS-1 leads to impaired insulin signaling and contributes to insulin resistance [13,14,15]. Signaling molecules such as c-Jun N-terminal kinase (JNK) [16,17], inhibitor of kappa B (IκB) kinase (IKK) [18], mammalian target of rapamycin (mTOR) [19,20], ribosomal protein S6 kinase (p70 S6K) [21,22,23], glycogen synthase kinase 3 (GSK3) [24], and protein kinase C (PKCs) [25,26] are involved in the serine phosphorylation of IRS-1 [12].

Adenosine monophosphate (AMP)-activated protein kinase (AMPK) is a serine/threonine kinase and a cellular energy sensor [27,28,29,30,31] that is activated by a reduced ATP/AMP ratio and/or via phosphorylation by its upstream kinases, liver kinase B1 (LKB1), calmodulin-dependent protein kinase (CaMKKs), and transforming growth factor-β (TGF-β)-activated kinase 1 (TAK1) [32,33]. Skeletal muscle AMPK is activated by muscle contraction/exercise [30], several antidiabetic drugs such as metformin [27], thiazolidinediones [34], and by naturally derived compounds/polyphenols such as resveratrol [35] and naringenin [36], resulting in increased glucose uptake. AMPK activators have been recognized, in recent years, as promising pharmacological interventions for the prevention and treatment of insulin resistance and T2DM [28,29,37,38,39].

According to the International Diabetes Federation (IDF) and the World Health Organization estimates, T2DM is a disease on the rise [40,41], posing a troublesome economic burden to health care systems globally. Although different strategies for the treatment of insulin resistance and T2DM currently exist, they are often lacking in efficacy and exhibit rare but potential life-threatening side-effects such as lactic acidosis, pancreatic cancer, pancreatitis, and a substantial decrease in quality of life [42]. Therefore, there is an urgent need for new preventative measures and targeted therapies. Chemicals/compounds found in plants/herbs have attracted attention, in recent years, for their use as nutraceuticals for preventing and treating insulin resistance and T2DM.

Rosemary (*Rosmarinus officinalis* L.) is an aromatic plant native to the Mediterranean region reported to exhibit potent antioxidant [43,44], anticancer [45,46], and antidiabetic properties [44,47,48,49,50,51,52]. Rosemary extract (RE) contains multiple classes of polyphenols including flavonoids, phenolic acids, and phenolic terpenes [51]. Carnosic acid (CA), carnosol (COH), and rosmarinic acid (RA) are the polyphenols found in the highest quantity in RE and their production is influenced by growth conditions such as water availability, soil quality, and sunlight exposure.

Previous studies by our group showed that the treatment of L6 muscle cells with RE [53] and CA [54] significantly increased glucose uptake and RE attenuated the palmitate-induced insulin resistance in L6 muscle cells [52]. Most importantly, CA robustly increased AMPK phosphorylation/activation in healthy muscle cells [54].

Studies have shown that the administration of CA or RE containing CA decreased plasma glucose levels in streptozotocin-induced diabetic rats [47,48,50,55], alloxan-induced diabetic rabbits [44], as well as genetic [49,55] and dietary [56,57,58,59,60] animal models of obesity and insulin resistance, indicating that the RE-derived polyphenols have the potential to be used as therapeutics against insulin resistance and T2DM. Other studies showed that treatment with CA attenuated the TNF-α-induced insulin resistance and nuclear kappa B (NF-κB)-mediated inflammation in 3T3-L1 adipocytes [61] and markedly suppressed adipogenesis in 3T3-L1 preadipocytes [62]. However, the effect of CA alone on palmitate-induced insulin-resistant muscle cells and adipocytes has never been investigated.

In the present study, we focused on CA and examined its potential to counteract the palmitate-induced insulin resistance in muscle and fat cells.

## 2. Materials and Methods

### 2.1. Materials

Fetal bovine serum (FBS), dimethyl sulfoxide (DMSO), palmitate, bovine serum albumin (BSA), carnosic acid, cytochalasin B, glutamine, 3-Isobutyl-1-methylxanthine (IBMX), dexamethasone, rosiglitazone, and anti-c-myc polyclonal antibody (CAT C3956) were purchased from Sigma Life Sciences (St. Louis, MO, USA). Materials for cell culture and trypan blue solution 0.4% were purchased from GIBCO Life Technologies (Burlington, ON, USA). Phospho—and total ACC (CAT 2661 and 3662, respectively), AMPK (CAT 2531 and 2532, respectively), Akt (CAT 9271 and 9272, respectively), JNK (CAT 9251 and 9252, respectively), mTOR (CAT 2971 and 2972, respectively), p70S6K (CAT 9205 and 9202, respectively) IRS-1 (CAT 2381, 2388, and 2382, respectively), and HRP-conjugated anti-rabbit antibodies (CAT 7074) were purchased from New England BioLabs (NEB) (Mississauga, ON, Canada). Phospho-IRS-1 Tyr^612^ (CAT 44-816G) was purchased from Invitrogen (Burlington, ON, Canada). Insulin (Humulin R) was from Eli Lilly (Indianapolis, IN, USA). Luminol Enhancer reagents, polyvinylidene difluoride (PVDF) membrane, reagents for electrophoresis, and Bradford protein assay reagent were purchased from BioRad (Hercules, CA, USA). [3H]-2-deoxy-D-glucose was purchased from PerkinElmer (Boston, MA, USA).

### 2.2. Preparation of Palmitate Stock Solution

Palmitate stock was prepared by the conjugation of palmitic acid with fatty acid-free BSA as previously described by us [52,63] and others [6]. Briefly, palmitic acid was dissolved in 0.1 N NaOH and was diluted in a previously pre-warmed (45–50 °C) BSA solution 9.7% (*w*/*v*). This process gave a stock solution of 8 mM palmitate with a final molar ratio of free palmitate/BSA of 6:1. The stock of palmitate was kept at −80 °C in small 1.5 mL Eppendorf tube aliquots.

### 2.3. Cell Culture, Treatment, and Glucose Uptake

For this study, we used L6 rat skeletal muscle cells (wild-type), GLUT4myc overexpressing L6 rat skeletal muscle cells, and 3T3-L1 adipocytes. Myoblasts were grown and differentiated into myotubes, as previously established by our lab and other groups [35,52,64]. Briefly, L6 muscle cells were grown in α-Minimum Essential Medium (MEM) media containing 2% (*v*/*v*) FBS until fully differentiated. Fully differentiated myotubes were used in all experiments. 3T3-L1 adipocytes were grown and propagated in Dulbecco’s Modified Eagle Medium (DMEM) containing 10% (*v*/*v*) FBS and 2 mM glutamine (basal media). After confluence was reached (approximately 4 days after seeding), the cells were exposed to differentiation induction DMEM medium (containing 0.5 mM IBMX, 0.25 μM dexamethasone, 1 μg/mL insulin, and 2 μM rosiglitazone) for 4 days followed by exposure to a second differentiation medium (DMEM containing 1 μg/mL insulin) for 4 days. All treatments were performed using serum-free media. The fully differentiated myotubes and 3T3L1 adipocytes were treated with 0.2 mM palmitate in the presence or absence of 2 µM CA for 16 h followed by treatment with 100 nM insulin for 30 min. After treatment, the cells were washed with HEPES-buffered saline (HBS) and exposed to HBS containing 10 µM [3H]-2-deoxy-D-glucose for 10 min to measure glucose uptake, as previously described [35,65]. At the end of the assay, the cells were washed with 0.9% NaCl solution (saline) and lysed using 0.05 N NaOH. The lysates were collected in scintillation vials containing scintillation fluid and their radioactivity was measured using a liquid scintillation β-counter (PerkinElmer).

### 2.4. GLUT4myc Translocation Assay

GLUT4myc overexpressing L6 myoblasts were seeded in 24-well plates grown until confluency and differentiated. After treatment, the cells were rinsed with ice-cold PBS and fixed with 3% paraformaldehyde (fixative) containing PBS for 10 min at 4 °C and placed on a rocker. The cells were then rinsed and incubated with 1% glycine containing PBS for 10 min at 4 °C followed by incubation with a blocking buffer of 10% goat serum containing-PBS for 15 min. The cells were then incubated with a blocking buffer containing primary anti-myc antibodies (1:500) for 1 h at 4 °C followed by thorough washing with PBS and incubation with HRP-conjugated donkey anti-mouse IgG-containing secondary antibody suspended in a blocking buffer (1:1000) for 45 min at 4 °C. Finally, the cells were rinsed with PBS and incubated with the O-phenylenediamine dihydrochloride (OPD) reagent for 30 min at room temperature and protected from light. The reaction was then stopped by the addition of 3 N HCl solution in each well. The supernatant was collected, and the absorbance was measured at 492 nm using a plate reader (Synergy HT, BioTek Instruments, Winooski, VT, USA). The color intensity is directly proportional to the GLUT4myc transporter levels present on the plasma membrane.

### 2.5. Immunoprecipitation of IRS-1

Whole-cell lysates (200 µg) were incubated with IRS-1 antibody (at a 1:50 volume ratio) and conjugated to SureBeads^TM^ Protein G Magnetic beads (Biorad; Hercules, CA, USA) for 1 h at room temperature followed by microcentrifugation and three washes with PBS + 0.1% Tween 20. Protein was eluted with a glycine (20 mM, pH 2.0) solution for 5 min at room temperature and neutralized with PBS (1 M, pH 7.4) at 10% eluent volume. A 3× SDS sample buffer was added to eluted protein and boiled for 5 min.

### 2.6. Immunoblotting

Following treatment, the cells were rinsed with ice-cold PBS and lysed using ice-cold lysis buffer containing 1 mM ethylenediaminetetraacetic acid (EDTA), 1 mM ethylene glycol-bis β-aminoethyl ether/egtazic acid (EGTA), 150 mM NaCI, 1% Triton X-100, 20 mM Tris (PH 7.5), 1mM sodium orthovanadate (Na_3_VO_4_), 1 µg/mL leupeptin, 2.5 mM sodium pyrophosphate, 1 mM p-glycerolphosphate, and 1 mM phenylmethylsulfonyl fluoride (PMSF). The lysates were collected in 1.5 mL Eppendorf tubes, 5% β-mercaptoethanol containing 3× SDS buffer was added, and the tubes were placed in boiling water for 5 min. Sodium dodecyl sulfate-polyacrylamide gel electrophoresis (SDS-PAGE) was used to separate proteins (samples of 20 µg), determined using the Bradford assay [66], followed by a transfer of the proteins to the PVDF membrane. The membranes were exposed to blocking buffer 5% (*w*/*v*) dry milk powder in Tris-buffered saline for 1 h, followed by overnight exposure at 4 °C to the primary antibody. The next day, the membrane was exposed to a blocking buffer containing secondary HRP-conjugated anti-rabbit antibody for 1 h followed by exposure to LumiGLOW reagents. The western blot bands were visualized using FluroChem software (Thermo Fisher, Waltham, MA, USA) and analyzed using ImageJ (imagej.nih.gov, accessed on January 2019–October 2021).

### 2.7. Statistical Analysis

Statistical analysis was performed using GraphPad Prism software 5.3 manufactured from Graphpad Software Inc. (La Jolla, CA, USA). The data from 3–7 experiments were pooled and presented as mean ± standard error (SE). Analysis of variance (ANOVA) followed by Tukey’s post hoc test for multiple comparisons, was used for statistical analysis.

## 3. Results

### 3.1. Carnosic Acid Restores the Insulin-Stimulated Glucose Uptake in Palmitate-Treated Muscle Cells and 3T3-L1 Adipocytes

The effects of the free fatty acid palmitate in the absence or the presence of CA on the insulin-stimulated glucose uptake were examined. Acute stimulation of L6 myotubes with insulin (100 nM, 30 min) significantly increased glucose uptake (I: 201 ± 1.21% of control, *p* < 0.0001, Figure 1A). Exposure of the cells to palmitate (0.2 mM, 16 h) did not have any effect on the basal glucose uptake (P: 105 ± 2.4% of control, Figure 1A) but significantly blunted the insulin-stimulated glucose uptake (P+I: 119 ± 13.2% of control, *p* < 0.0001), indicating insulin resistance. Most importantly in palmitate-treated cells, exposure to CA (2 µM) resulted in significant restoration of insulin-stimulated glucose uptake (CA+P+I: 179 ± 10.5% of control *p* < 0.0001, Figure 1A). These data indicate that the negative effect imposed by palmitate treatment on insulin responsiveness is abolished in the presence of CA. CA treatment alone significantly increased the basal (CA: 253 ± 19.8% of control, *p* < 0.01 vs. control) and the insulin-stimulated glucose uptake (CA+I: 269 ± 13.2% of control, *p* < 0.001 vs. control, *p* < 0.05 vs. insulin, Figure 1A insert).

Acute stimulation of fully differentiated 3T3-L1 adipocytes with insulin (100 nM, 30 min) resulted in a significant increase in glucose uptake (I: 151 ± 9.70% of control, *p* < 0.01, Figure 1B). Palmitate abolished the insulin-stimulated glucose uptake (P+I: 102 ± 9.90% of control, *p* < 0.01) while treatment with CA improved the insulin response (CA+P+I: 139 ± 7.61% of control, *p* < 0.01, Figure 1B). Treatment with CA significantly increased the basal (CA: 152 ± 4.5% of control, *p* < 0.001) but not the insulin-stimulated glucose uptake (CA+I: 166 ± 2.4% of control, *p* < 0.001, Figure 1B insert).

### 3.2. Carnosic Acid Restores the Insulin-Stimulated GLUT4 Translocation in Palmitate Treated Myotubes

The increase in muscle glucose uptake seen with acute insulin stimulation is due to the translocation of the GLUT4 glucose transporter from an intracellular pool to the plasma membrane. To examine the effects of our treatment on GLUT4, we used L6 cells that overexpress a myc-labeled GLUT4 glucose transporter. Acute stimulation of GLUT4myc overexpressing L6 myotubes with insulin (100 nM, 30 min) resulted in a significant increase in GLUT4 plasma membrane levels (I: 188.7 ± 10.8% of control, *p* < 0.01, Figure 2). Palmitate significantly reduced the acute insulin-mediated increase in GLUT4 plasma membrane levels, indicating insulin resistance (P+I: 135.2 ± 5.8% of control, *p* < 0.05, Figure 2); this response was restored in the presence of CA (CA+P+I: 194.4 ± 11.4% of control, *p* < 0.05, Figure 2). Treatment with CA alone did not affect basal (CA: 109.4 ± 3.4% of control, *p* < 0.05) or insulin-stimulated GLUT4 plasma membrane levels (CA+I: 196.2 ± 7.9% of control) (Figure 2 insert), as seen previously [54].

### 3.3. Carnosic Acid Restores the Insulin-Stimulated Akt Phosphorylation in Palmitate Treated Myotubes

Next, we investigated the effect of palmitate and CA treatment on insulin stimulated Akt phosphorylation and expression. Treatment of L6 myotubes with insulin resulted in a significant increase in Akt (Ser^473^ residue) phosphorylation (I: 381.5 ± 58.76% of control, *p* = 0.0008, Figure 3A,B). Treatment of the cells with palmitate reduced the insulin-stimulated Akt phosphorylation (P+I: 138.1 ± 26.9% of control, *p* = 0.009, Figure 3A,B). However, in the presence of CA, the decline in the insulin-stimulated Akt phosphorylation seen with palmitate was completely prevented (CA+P+I: 335.6 ± 26.3% of control, *p* < 0.01, Figure 3A,B). Palmitate alone did not have a significant effect on the basal Akt phosphorylation (P: 77.8 ± 11.6% of control, Figure 3A,B). The total levels of Akt were not significantly affected by any of the treatments (I: 104.3 ± 12.2, P: 92.4 ± 4.8, P+I: 86.1 ± 4.02, CA+P+I: 85.3 ± 6.9% of control, Figure 3A).

### 3.4. Carnosic Acid Prevents the Palmitate-Induced Serine Phosphorylation of IRS-1 in L6 Myotubes

Increased serine phosphorylation of IRS-1 is linked to insulin resistance [15,67,68,69] and for this reason, we examined IRS-1. Exposure of the cells to palmitate (0.2 mM, 16 h) significantly increased IRS-1 phosphorylation at residue Ser307 and Ser636/639 (P: 136.8 ± 5.24% and 138.4 ± 9.32% of control, respectively, both *p* < 0.01, Figure 4A,B) and treatment with CA completely abolished the palmitate-induced Ser307 and Ser636/639 phosphorylation of IRS-1 (CA+P: 107.8 ± 2.64% of control, *p* < 0.01, and 103.1 ± 8.67% of control, *p* < 0.05, respectively, Figure 4A,B). CA alone did not affect Ser307 and Ser636/639 phosphorylation of IRS-1 (CA: 95.5 ± 7.00% of control, and 100.8 ± 11.57% of control, respectively, Figure 4A,B). Moreover, the total levels of IRS-1 were not significantly changed by any treatment (P: 106.3 ± 3.18%, CA: 101.5 ± 0.93%, CA+P: 102.3 ± 1.03% of control, Figure 4A).

### 3.5. Carnosic Acid Prevents the Palmitate-Induced Phosphorylation of C-Jun N-Terminal Kinase (JNK) in L6 Myotubes

Following the evidence that exposure to palmitate increases the phosphorylation of serine residues of IRS-1, we examined the signaling molecules that may be involved. JNK is a serine/threonine kinase shown to increase serine phosphorylation of IRS-1 and is involved in insulin resistance [70,71]. We hypothesized that the levels of JNK phosphorylation and/or expression would be increased by palmitate. Indeed, exposure of the cells to palmitate (0.2 mM, 16 h) significantly increased JNK phosphorylation (P: 149.9 ± 2.29% of control, *p* < 0.0001, Figure 5A,B) and treatment with CA completely abolished the palmitate-induced phosphorylation of JNK (CA+P: 62.73 ± 7.34% of control, *p* < 0.001, Figure 5A,B). CA alone significantly reduced the phosphorylation of JNK (CA: 68.95 ± 1.06% of control, *p* < 0.0001, Figure 5A,B). The total levels of JNK were not significantly changed by any treatment (P: 104 ± 5.32%, CA: 90 ± 13.3%, CA+P: 122 ± 17.9% of control, Figure 5).

### 3.6. Carnosic Acid Prevents the Palmitate-Induced Phosphorylation of mTOR and p70S6K in L6 Myotubes

Additional kinases implicated in serine phosphorylation of IRS-1 are mTOR and p70S6K, and we examined the effects of palmitate and CA on these kinases. Exposure of the cells to 0.2 mM palmitate for 16 h significantly increased mTOR Ser^2448^ and p70S6K Thr^389^ phosphorylation (P: 166 ± 14.39% of control and 141.3 ± 2.8% of control respectively, both *p* < 0.01, Figure 6A,B). Treatment with CA had no effect on the basal mTOR phosphorylation (CA: 74.89 ± 11.02% of control, *p* < 0.001, Figure 6A,B) and completely abolished the palmitate-induced phosphorylation of mTOR (CA+P: 91.47 ± 11.65% of control, *p* = 0.001, Figure 6A,B). Similarly, treatment with CA significantly reduced the palmitate-induced phosphorylation of p70S6K (CA+P: 104.9 ± 2.9% of control, *p* < 0.05, Figure 6A,B). The total levels of mTOR and p70S6K were not significantly changed by any treatment (P: 93 ± 8.36% and 101 ± 10.2%, CA: 102 ± 8.43% and 86.7 ± 4.8%, CA+P: 109 ± 13.8% and 95 ± 8.5% of control, respectively, Figure 6A,B).

The activity of mTOR is influenced by the raptor (regulatory-associated protein) of the mammalian target of rapamycin (mTOR). Phosphorylation of raptor on Ser^792^ inhibits mTOR [72,73,74]. We examined both phosphorylated (Ser^792^) and total raptor. Treatment with CA alone (CA: 169% of control) or in the presence of palmitate (CA+P: 239% of P) increased raptor phosphorylation (Figure 6C).

### 3.7. Carnosic Acid Increases the Phosphorylation of AMPK and ACC in the Presence of Palmitate in L6 Myotubes

Previous studies by our group showed that rosemary extract and rosemary extract polyphenols acutely increased glucose uptake and phosphorylated/activated AMPK in L6 muscle cells [52,53,54,75,76]. Here we investigated the prolonged effect of CA on AMPK and its downstream target ACC under basal and elevated FFA conditions. Treatment with 2 μM CA increased the phosphorylation of AMPK and ACC (CA: 145.2 ± 11.01% and 145.8 ± 2.01% of control, respectively, Figure 7A,B). Most importantly, CA increased the phosphorylation of AMPK (CA+P: 151.6 ± 12.7% of control, *p* < 0.05) and ACC (CA+P: 144.8 ± 14.01% of control, *p* < 0.05), even in the presence of palmitate (Figure 7A,B). Treatment with palmitate alone had no significant effect on the phosphorylation of AMPK and ACC (P: 109% and 82% of control, respectively, Figure 7A,B). Furthermore, the total levels of AMPK or ACC were not affected by any treatment (P: 92% and 107%, CA: 92% and 104%, CA+P: 90% and 99% of control, respectively, Figure 7).

### 3.8. Carnosic Acid Prevents the Effects of Palmitate in 3T3-L1 Adipocytes

Similar to the experiments we conducted in L6 myotubes, we examined the effect of CA on signaling molecules in palmitate-treated 3T3-L1 adipocytes. Treatment of 3T3-L1 adipocytes with insulin resulted in a significant increase in Akt (Ser^473^ residue) phosphorylation (I: 225.5 ± 33.16% of control, *p* = 0.0387, Figure 8A) and this response was abolished in the presence of palmitate (P+I: 73.35 ± 43.77% of control, *p* = 0.0155, Figure 8A). However, in the presence of CA, the decline in the insulin stimulated Akt phosphorylation seen with palmitate was completely prevented (CA+P+I: 272.8 ± 42.01% of control, *p* < 0.01, Figure 8A). Palmitate alone did not have a significant effect on the basal Akt phosphorylation. The total levels of Akt were not significantly affected by any of the treatments (I: 108.6 ± 11.19%, P: 114 ± 11.66%, P+I: 102.7 ± 11.75%, CA+P+I: 103.9 ± 13.31% of control, Figure 8A).

Exposure of the cells to palmitate (0.4 mM, 16 h) significantly increased serine phosphorylation of IRS-1 (P: 151 ± 18.89% of control, *p* < 0.01, Figure 8B) and treatment with CA abolished this palmitate-induced effect (CA+P: 54.75 ± 9.56% of control, *p* < 0.01, Figure 8B). CA alone did not affect the phosphorylation of IRS-1, and the total levels of IRS-1 were not significantly changed by any treatment.

Exposure of the cells to palmitate significantly increased JNK (P: 205.2 ± 21.34% of control, *p* < 0.05, Figure 9A), mTOR (P: 209.2 ± 20.5% of control, *p* = 0.002, Figure 9B), and p70S6K (P: 259.4 ± 24.33% of control, *p* < 0.0001, Figure 9C) phosphorylation. Treatment with CA completely abolished the palmitate-induced phosphorylation of JNK (CA+P: 87.25 ± 29.20% of control, *p* < 0.05, Figure 9A), mTOR (CA+P: 97.41 ± 26.22% of control, *p* = 0.0016, Figure 9B), and p70S6K (CA+P: 98.96 ± 3.252% of control, *p* < 0.0001, Figure 9C). Treatment with CA alone did not have an effect on the basal JNK (CA: 110.1 ± 12.78% of control, *p* < 0.0001, Figure 9A), mTOR (CA: 110.7 ± 9.618% of control, *p* = 0.97, Figure 9B), or p70S6K (CA: 124.8 ± 5.623% of control, *p* < 0.0001, Figure 9C) phosphorylation. The total levels of JNK (Figure 9A), mTOR (Figure 9B), and p70S6K (Figure 9C) were not significantly changed by any treatment.

We also investigated the effect of CA on adipocyte AMPK and ACC. We performed a dose-response experiment; to the best of our knowledge, such experiments with CA were not previously performed in adipocytes. Treatment of 3T3-L1 adipocytes with 2, 10, or 20 μM CA for 16 h resulted in a dose-dependent increase in the phosphorylation/activation of AMPK (CA 2 μM: 140.3 ± 12.77%, CA 10 μM: 263.2 ± 41.12%, CA 20 μM: 410.2 ± 97.98%, Figure 10A) and ACC (CA 2 μM: 141.3 ± 11.42%, CA 10 μM: 229.8 ± 13.58%, CA 20 μM: 240 ± 33.64% of control, Figure 10B). Importantly, CA (20 μM) increased the phosphorylation of AMPK (CA+P: 425.9 ± 141.6% of control, *p* < 0.05) and ACC (CA+P: 253.5 ± 42.07% of control, *p* < 0.01), even in the presence of palmitate (Figure 10A,B). Treatment with palmitate alone had no significant effect on the phosphorylation of AMPK or ACC (P: 143 ± 28.91% and 128.2 ± 35.05% of control, respectively, Figure 10A,B). Furthermore, the total levels of AMPK or ACC were not affected by any treatment (P: 85.70 ± 13.17% and 107.6 ± 37.61%, CA: 79.05 ± 12.58% and 133.8 ± 33.31%, CA+P: 82.30 ± 18.34% and 102.8 ± 17.29% of control, respectively, Figure 10A,B).

## 4. Discussion

Elevated FFAs and obesity are major risk factors for the development of insulin resistance and T2DM [4,12], a disease on the rise, affecting millions of people globally. Finding compounds with the potential to counteract insulin resistance will provide huge benefits worldwide.

In the present study, we found that exposure of L6 myotubes to palmitate, to mimic the in vivo elevated plasma FFA levels seen in obesity, significantly decreased the insulin-stimulated glucose uptake and reduced the plasma membrane levels of the glucose transporter GLUT4, indicating the induction of insulin resistance. In agreement with these data, previous studies by our group and others found that the exposure of skeletal muscle cells to similar concentrations of palmitate induced insulin resistance [6,63,77,78,79]. Importantly, we found that in the presence of CA, the palmitate-induced insulin resistance was prevented, and the insulin-stimulated glucose uptake and GLUT4 plasma membrane levels were restored to levels comparable to the response seen with insulin alone. In addition, palmitate-induced adipocyte insulin resistance was prevented in the presence of CA. These findings are the first to show that CA can counteract the palmitate-induced muscle and adipocyte insulin resistance. CA alone increased glucose uptake in L6 muscle cells (Figure 1A insert) without affecting GLUT4 plasma membrane levels (Figure 2 insert) in agreement with the previous finding by our group [54], indicating that at the basal level, CA may influence the intrinsic activity of plasma membrane glucose transporters.

Furthermore, we found that exposure of the cells to palmitate significantly attenuated the insulin-stimulated phosphorylation of Akt in L6 and 3T3-L1 cells. These data agree with other in vitro studies using L6 [63,80], C2C12 [81] cells, and 3T3-L1 adipocytes [82], as well as in vivo studies that showed an attenuation of the insulin-stimulated phosphorylation of Akt in the soleus muscle isolated from animals fed a high-fat diet [83]. Interestingly, in our study, in the presence of CA, the insulin-stimulated phosphorylation of Akt was restored, suggesting that CA has the potential to counteract the deleterious effects of palmitate. These effects are similar to metformin, which has been shown to counteract the effects of palmitate and restore insulin-stimulated Akt phosphorylation in L6 muscle cells [84].

Exposure of L6 muscle cells and 3T3-L1 adipocytes to palmitate significantly increased the phosphorylation of JNK in agreement with other studies in L6 [85], C2C12 [86] muscle cells, 3T3-L1 adipocytes [82], and in muscle tissue from animals fed a high-fat diet [70,87]. Our data show that treatment with CA prevented the palmitate-induced phosphorylation of JNK. These data are in agreement with a study that showed that treatment of L6 muscle cells and muscles obtained from *ob/ob* mice with quercetin, a polyphenol from the flavonoid group, significantly attenuated the palmitate-induced phosphorylation of JNK [85]. In addition, exposure of L6 and 3T3-L1 adipocytes to palmitate significantly increased the phosphorylation of mTOR and p70S6K, and treatment with CA abolished these palmitate effects. Although increased mTOR phosphorylation by palmitate has been reported previously in L6 [52,64,85], C2C12 cells [88,89], 3T3-L1 adipocytes [90] and in muscle tissue from animals fed a high-fat diet [85,88,89], our study is the first to show that CA has the potential to block these effects. Our data show an effect of CA similar to metformin [89] and rapamycin [91].

We also investigated the phosphorylated and total levels of AMPK, ACC, and raptor. Previously, we found that treatment of L6 myotubes with RE [52,53] and the RE polyphenols CA [54], RA [75], and carnosol [76] significantly increased the phosphorylation of AMPK. In the present study, we found that palmitate (0.2 mM; 16 h) had no effect on AMPK or ACC. Treatment with CA increased the phosphorylation of AMPK and its downstream target ACC even in the presence of palmitate. The effect of CA is similar to metformin, which has been shown to phosphorylate/activate AMPK in the presence of palmitate in C2C12 and L6 muscle cells [84,89]. Several studies have indicated that activation of AMPK significantly lowers the activity of mTOR and its downstream effector p70 S6K [30,37,91,92] and, therefore, the inhibition of mTOR phosphorylation by CA treatment seen in our study may be mediated by AMPK activation. To elucidate the mechanistic link between AMPK activation and mTOR inhibition by CA we examined raptor. Raptor is a constitutively binding protein of mTOR complex 1 (mTORC1) and plays a role in regulating mTOR activity. Importantly, activated AMPK directly phosphorylates raptor on Ser^722/792^ [72], leading to inhibition of mTOR activity. Treatment with CA increased AMPK phosphorylation/activation and increased raptor phosphorylation, suggesting that the inhibition of mTOR phosphorylation/activation is mediated by AMPK activation. Future studies using strategies to inhibit AMPK such as using small interference RNA (siRNA) techniques or an inhibitor of AMPK (Compound C) should be performed to explore this further.

A limited number of studies have also examined the anti-diabetic effects of CA in vivo. The administration of RE enriched with CA significantly ameliorated the high-fat diet-induced obesity and insulin resistance in mice [59]. The administration of RE enriched with CA in obese rats resulted in a significant reduction of circulating plasma TNFα and interleukin-1α (IL-1α) levels indicating anti-inflammatory effects of CA [92]. Dietary supplementation of RE enriched with CA resulted in body weight and fat reduction [93] and inhibition of hepatic steatosis [58]. In another study, co-administration of diabetic rats with CA and RA reduced fasting plasma glucose, total cholesterol, and triglyceride levels and attenuated the formation of malondialdehyde (a marker of oxidative stress) and pro-inflammatory cytokine protein levels [94]. A major complication of diabetes mellitus, diabetic nephropathy, was attenuated in mice administered CA, with reduced albuminuria, and improved kidney function [95]. These studies demonstrate that the RE polyphenol CA exhibits antihyperglycemic and anti-diabetic properties in vivo. However, there are currently no studies that elucidate the mechanism involved in the effects of CA. The present study found that CA activated AMPK, reduced the palmitate-induced phosphorylation of mTOR, p70S6K, and JNK, and restored the insulin stimulated Akt phosphorylation, plasma membrane GLUT4 levels, and glucose uptake.

## 5. Conclusions

According to the International Diabetes Federation, the prevalence of T2DM is constantly increasing and is expected to affect 420 million people worldwide by the year 2040 [40]. Insulin resistance and T2DM increase the risk of cardiovascular disease and cancer [96,97]. Therefore, new strategies to aid in the prevention and management of T2DM will provide huge benefits to our society. Increased levels of FFA mediate insulin resistance in muscle and fat cells. In the present study, exposure of muscle and fat cells to the FFA palmitate, to mimic the elevated FFA levels seen in obesity, induced insulin resistance. Palmitate increased serine phosphorylation of IRS-1 and the phosphorylation/activation of JNK, mTOR, and p70S6K, while the insulin-stimulated Akt phosphorylation and the insulin-stimulated glucose uptake and GLUT4 translocation were significantly reduced. Importantly, these effects of palmitate were attenuated by CA and the insulin-stimulated glucose uptake was restored (Figure 11). In addition, CA increased the phosphorylation/activation of the energy sensor AMPK and increased phosphorylation of its downstream targets ACC and raptor. Our study is the first to show that CA has the potential to counteract the palmitate-induced muscle and fat cell insulin resistance and further studies are required to explore its anti-diabetic properties and to elucidate the exact cellular mechanisms involved.

## Figures and Tables

**Figure 1 cells-11-00167-f001:**
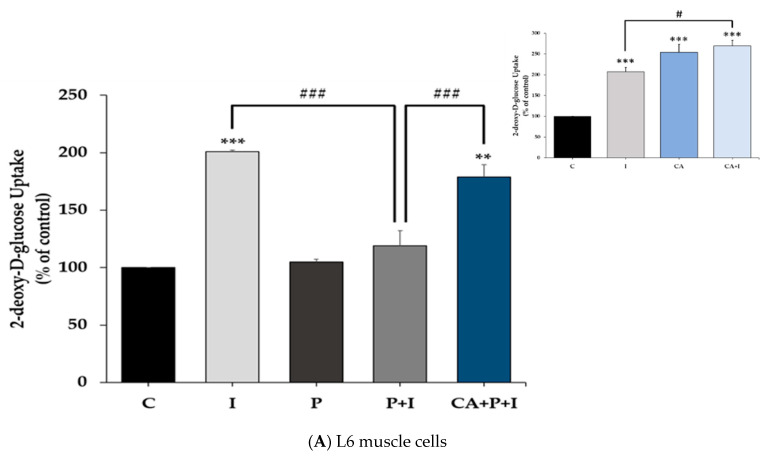
Carnosic acid restores the insulin-stimulated glucose uptake in palmitate treated skeletal muscle (**A**) and fat cells (**B**). Fully differentiated L6 myotubes (**A**) and 3T3-L1 adipocytes (**B**) were treated with 0.2 mM palmitate (P) for 16 h in the absence or the presence of 2 µM carnosic acid (CA) followed by stimulation without or with 100 nM insulin (I) for 30 min and [3H]-2-deoxy-D-glucose uptake measurements. The results are the mean ± standard error (SE) of four to six independent experiments, expressed as percent of control (* *p* < 0.05, ** *p* < 0.01, *** *p* < 0.001 vs. control, ### *p* < 0.001, ## *p* < 0.01, # *p* < 0.05 as indicated).

**Figure 2 cells-11-00167-f002:**
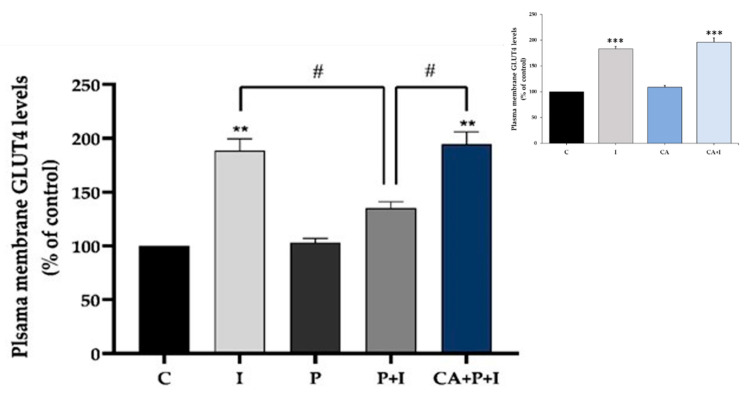
Effects of palmitate and carnosic acid on GLUT4 translocation in skeletal muscle cells. GLUT4myc overexpressing L6 myotubes were treated without (control, C) or with 0.2 mM palmitate (P) for 16 h in the absence or the presence of 2 μM carnosic acid (CA) followed by washing, as indicated in the methods, and acute stimulation with 100 nM insulin for 30 min (I). After treatment, plasma membrane GLUT4 glucose transporter levels were measured. Results are the mean ± SE of three independent experiments performed in triplicate (** *p* < 0.01, *** *p* < 0.001 vs. control, # *p* < 0.05 as indicated).

**Figure 3 cells-11-00167-f003:**
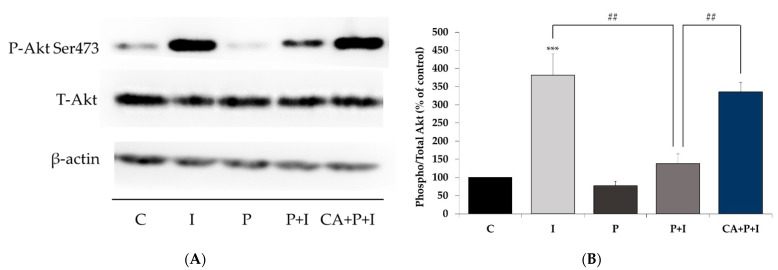
Effects of palmitate and carnosic acid on skeletal muscle cell Akt expression and phosphorylation/activation. Fully differentiated L6 myotubes were treated without (control, C) or with 0.2 mM palmitate (P) for 16 h in the absence or the presence of 2 μM carnosic acid (CA) followed by stimulation without or with 100 nM insulin (I) for 30 min. After treatment, the cells were lysed, and SDS-PAGE was performed, followed by immunoblotting with specific antibodies that recognize phosphorylated (Ser^473^), total Akt, or β-actin. Representative immunoblots are shown (**A**). The densitometry of the bands was measured and expressed in arbitrary units (**B**). The data are the mean ± SE of three to five separate experiments (*** *p* < 0.001 vs. control, ## *p* < 0.01 as indicated).

**Figure 4 cells-11-00167-f004:**
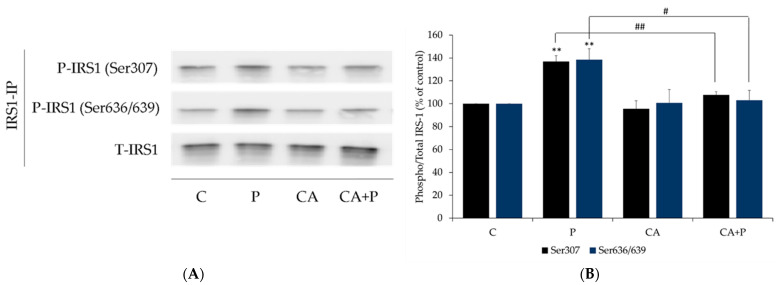
Effects of palmitate and carnosic acid on serine phosphorylation and expression of skeletal muscle cell IRS-l. Fully differentiated myotubes were treated without (control, C) or with 0.2 mM palmitate (P) for 16 h in the absence or presence of 2 µM carnosic acid (CA). After treatment, the cells were lysed, and IRS-1 immunoprecipitation was performed, followed by SDS-PAGE and immunoblotting with specific antibodies that recognize phosphorylated Ser^307^, Ser^636/639^, or total IRS-1. Representative immunoblots are shown (**A**). The densitometry of the bands was measured and expressed in arbitrary units (**B**). The data is the mean ± SE of three separate experiments. (** *p* < 0.01 vs. control, # *p < 0.05,* ## *p* < 0.01 vs. palmitate alone).

**Figure 5 cells-11-00167-f005:**
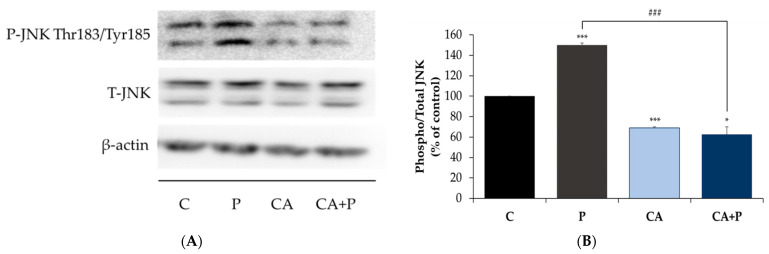
Effects of palmitate and carnosic acid on JNK expression and phosphorylation in skeletal-muscle cells. Fully differentiated L6 myotubes were treated without (control, C) or with 0.2 mM palmitate (P) for 16 h in the absence or the presence of 2 μM carnosic acid (CA). After treatment, the cells were lysed, and SDS-PAGE was performed, followed by immunoblotting with specific antibodies that recognize phosphorylated Thr^183^/Tyr^185^ or total JNK. Representative immunoblots are shown (**A**). The densitometry of the bands was measured and expressed in arbitrary units (**B**). The data are the mean ± SE of three separate experiments (* *p* < 0.05, *** *p* < 0.001 vs. control, ### *p* < 0.001 vs. palmitate alone).

**Figure 6 cells-11-00167-f006:**
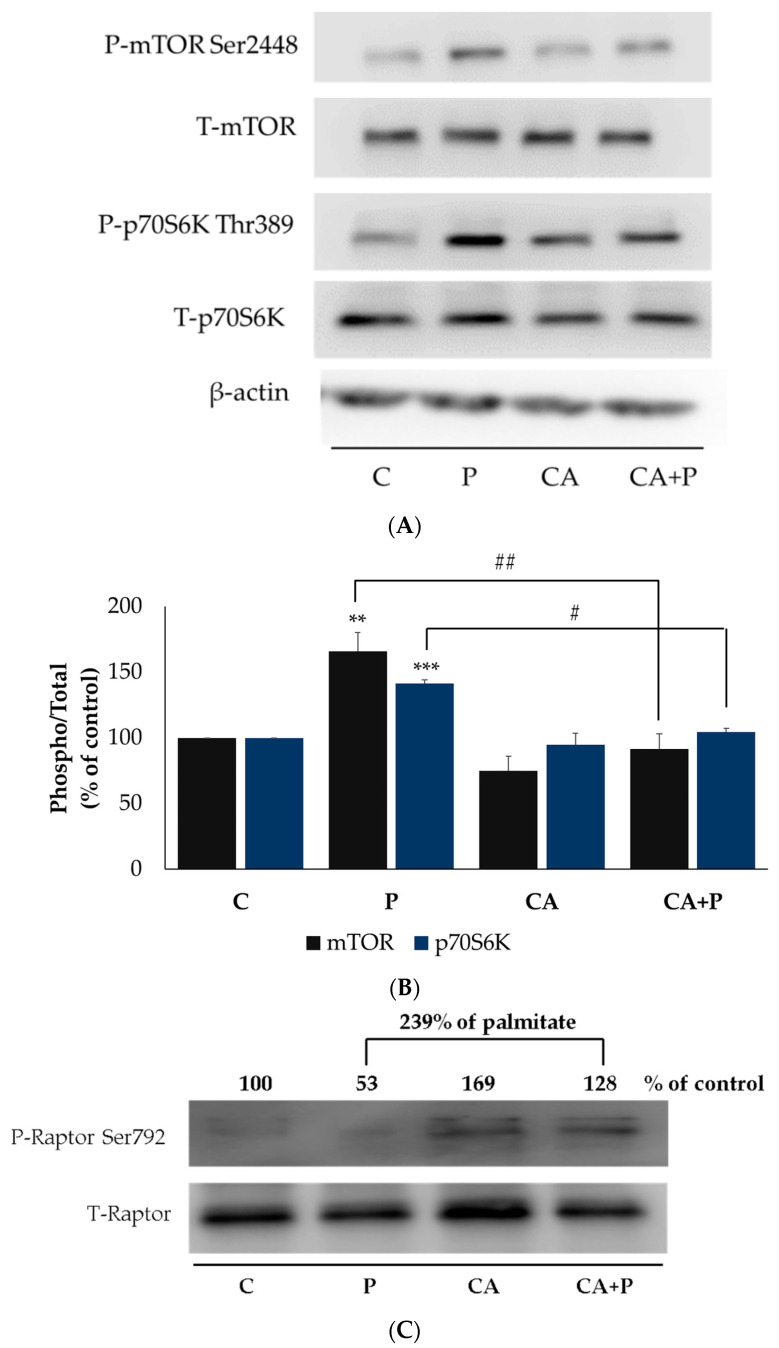
Effects of palmitate and carnosic acid on mTOR, p70S6K and Raptor expression and phosphorylation in skeletal muscle cells. Fully differentiated myotubes were treated without (control, C) or with 0.2 mM palmitate (P) for 16 h in the absence or the presence of 2 μM carnosic acid (CA). After treatment, the cells were lysed, and SDS-PAGE was performed, followed by immunoblotting with specific antibodies that recognize phosphorylated Ser^2448^ or total mTOR, phosphorylated Thr^389^ or total p70S6K (**A**,**B**), and phosphorylated Ser^792^ or total Raptor (**C**). Representative immunoblots are shown (**A**,**C**). The densitometry of the bands was measured and expressed in arbitrary units (**B**). The data are the mean ± SE of four separate experiments (** *p < 0.01,* *** *p* < 0.001 vs. control, # *p* < 0.05, ## *p* < 0.01 vs. palmitate alone).

**Figure 7 cells-11-00167-f007:**
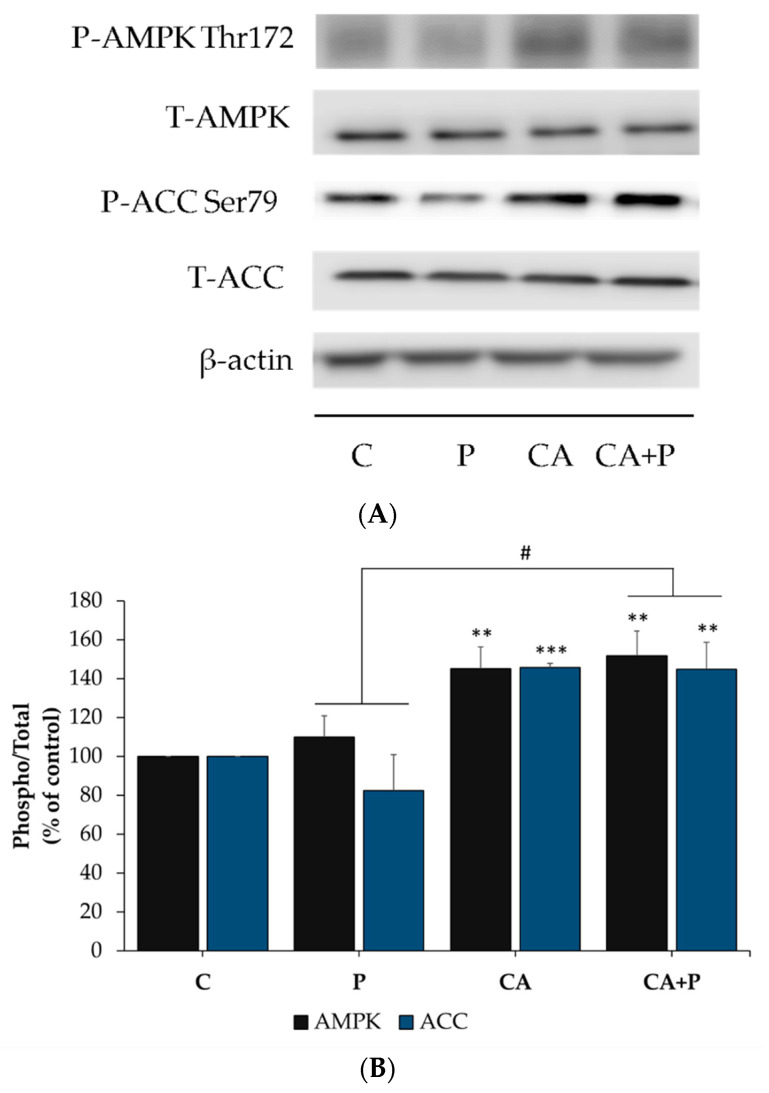
Effects of palmitate and carnosic acid on AMPK and ACC expression and phosphorylation in skeletal muscle cells. Fully differentiated myotubes were treated without (control, C) or with 0.2 mM palmitate (P) for 16 h in the absence or the presence of 2 μM carnosic acid (CA). After treatment, the cells were lysed, and SDS-PAGE was performed, followed by immunoblotting with specific antibodies that recognize phosphorylated Thr^172^, total AMPK, phosphorylated Ser^79^, or total ACC. Representative immunoblots are shown (**A**). The densitometry of the bands was measured and expressed in arbitrary units (**B**). The data is the mean ± SE of three separate experiments. (** *p* < 0.01, *** *p* < 0.001 vs. control, # *p* < 0.05 vs. palmitate alone).

**Figure 8 cells-11-00167-f008:**
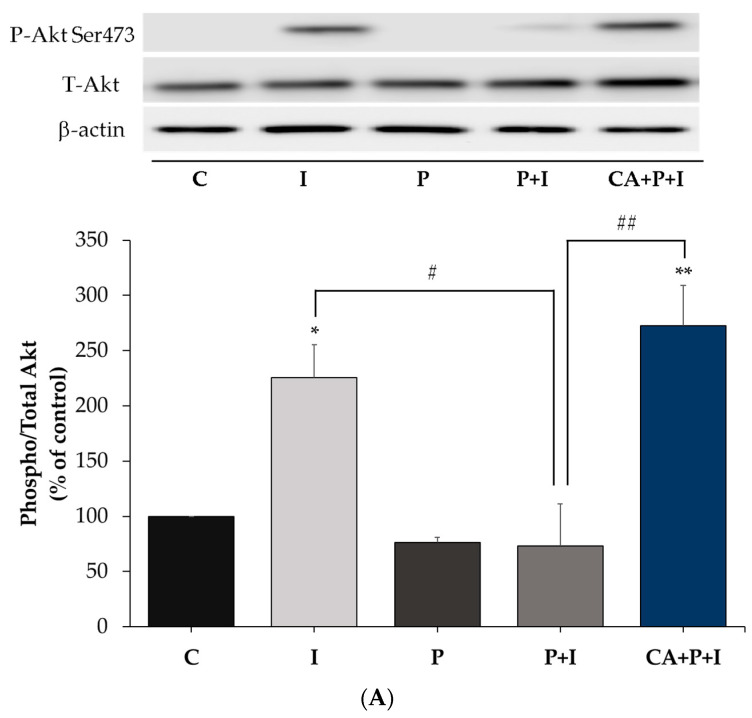
Effects of palmitate and carnosic acid on Akt expression and phosphorylation/activation and serine phosphorylation and expression of IRS-l in adipocytes. Fully differentiated 3T3-L1 adipocytes were treated without (control, C) or with 0.4 mM palmitate (P) for 16 h in the absence or the presence of 20 μM carnosic acid (CA) followed by stimulation without or with 100 nM insulin (I) for 30 min. After treatment, the cells were lysed, and SDS-PAGE was performed, followed by immunoblotting with specific antibodies that recognize phosphorylated Ser^473^ or total Akt (**A**) and Ser^307^ or total IRS-1 (**B**). Representative immunoblots are shown. The densitometry of the bands was measured and expressed in arbitrary units. The data are the mean ± SE of four separate experiments (* *p* < 0.05, ** *p* < 0.01 vs. control, # *p* < 0.05, ## *p* < 0.01 as indicated).

**Figure 9 cells-11-00167-f009:**
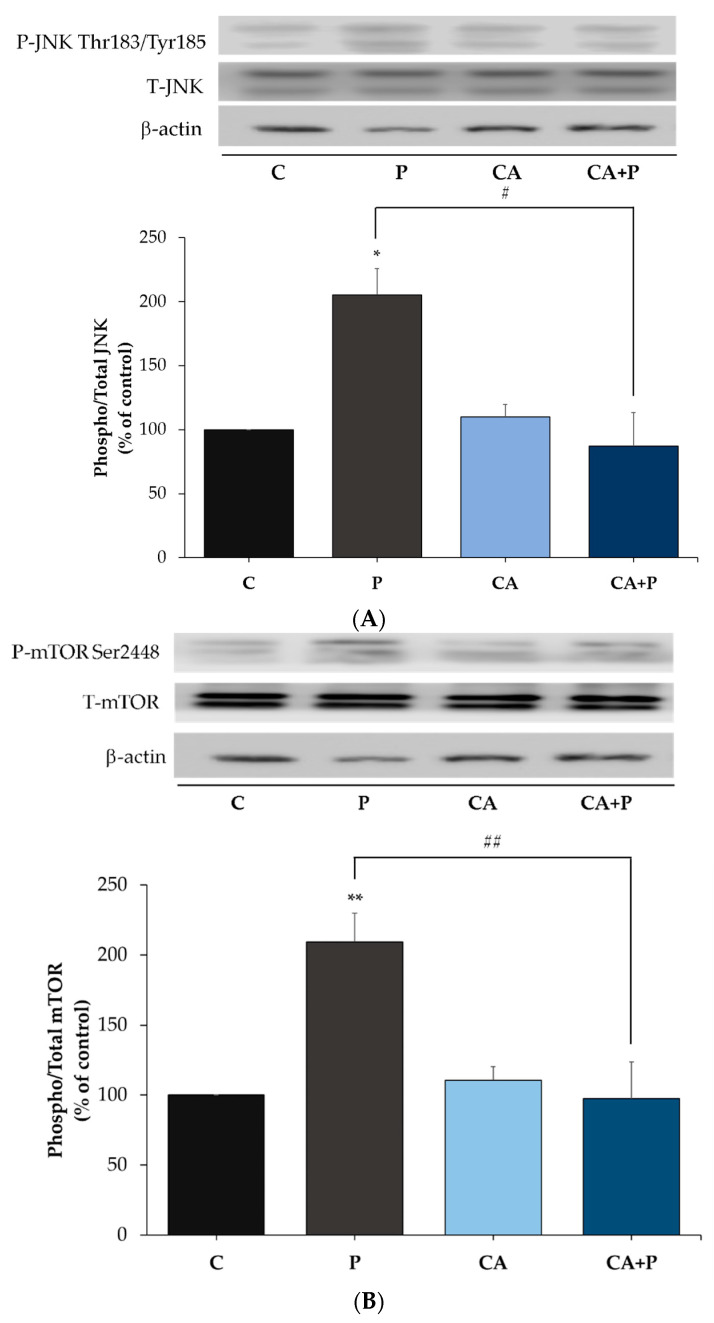
Effects of palmitate and carnosic acid on JNK, mTOR, and p70S6K expression and phosphorylation in adipocytes. Fully differentiated adipocytes were treated without (control, C) or with 0.4 mM palmitate (P) for 16 h in the absence or the presence of 20 μM carnosic acid (CA). After treatment, the cells were lysed, and SDS-PAGE was performed, followed by immunoblotting with specific antibodies that recognize phosphorylated Thr^183^/Tyr^185^ or total JNK (**A**), phosphorylated Ser^2448^ or total mTOR (**B**), and phosphorylated Thr^389^, or total p70S6K (**C**). Representative immunoblots are shown. The densitometry of the bands was measured and expressed in arbitrary units. The data are the mean ± SE of four separate experiments (* *p* < 0.05, ** *p* < 0.01, *** *p* < 0.001 vs. control, # *p* < 0.05, ## *p* < 0.01, ### *p* < 0.001 vs. palmitate alone).

**Figure 10 cells-11-00167-f010:**
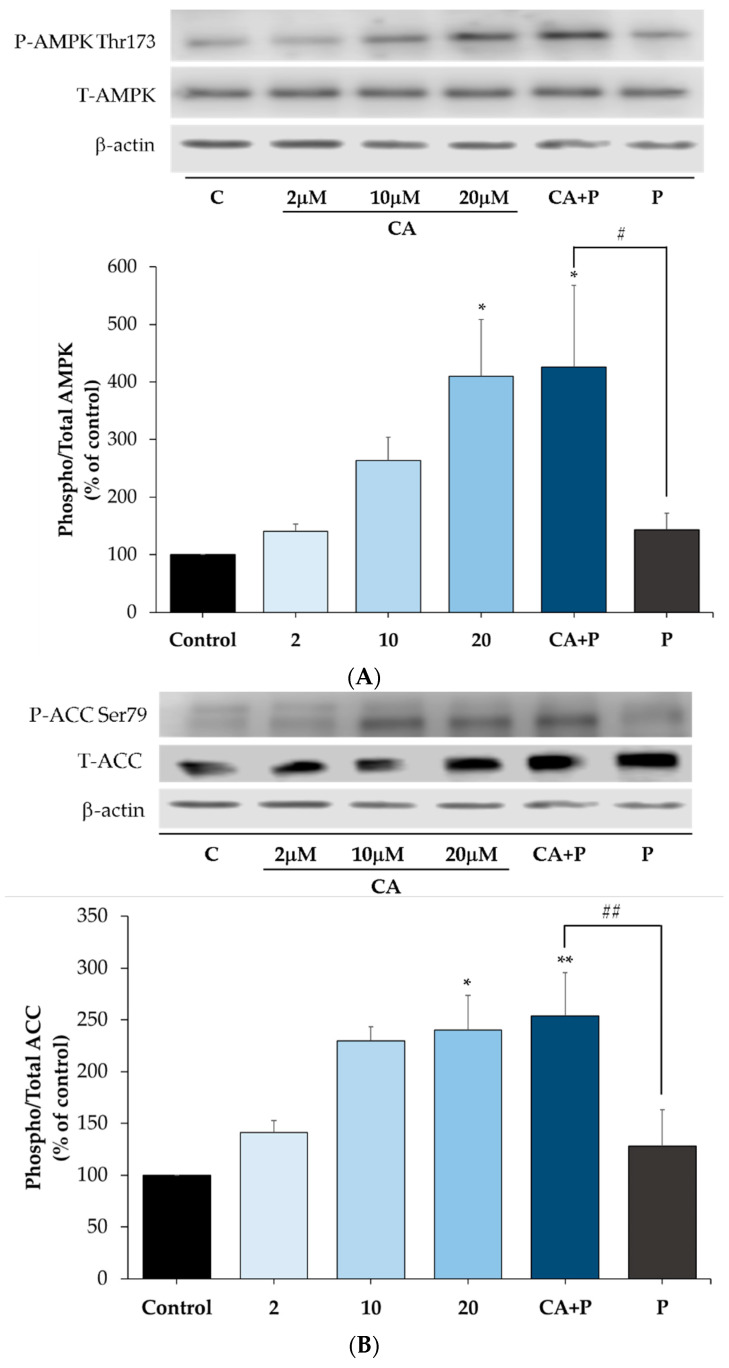
Effects of palmitate and carnosic acid on AMPK expression and phosphorylation in adipocytes. Fully differentiated adipocytes were treated without (control, C) or with 0.4 mM palmitate (P) for 16 h in the absence or the presence of 20 μM carnosic acid (CA). After treatment, the cells were lysed, and SDS-PAGE was performed, followed by immunoblotting with specific antibodies that recognize phosphorylated Thr^172^ or total AMPK and Ser^79^ or total ACC. Representative immunoblots are shown (**A**). The densitometry of the bands was measured and expressed in arbitrary units (* *p* < 0.05 vs. control, # *p* < 0.05 vs. palmitate alone). (**B**) The data is the mean ± SE of three separate experiments. (* *p* < 0.05, ** *p* < 0.01 vs. control, ## *p* < 0.01 vs. palmitate alone).

**Figure 11 cells-11-00167-f011:**
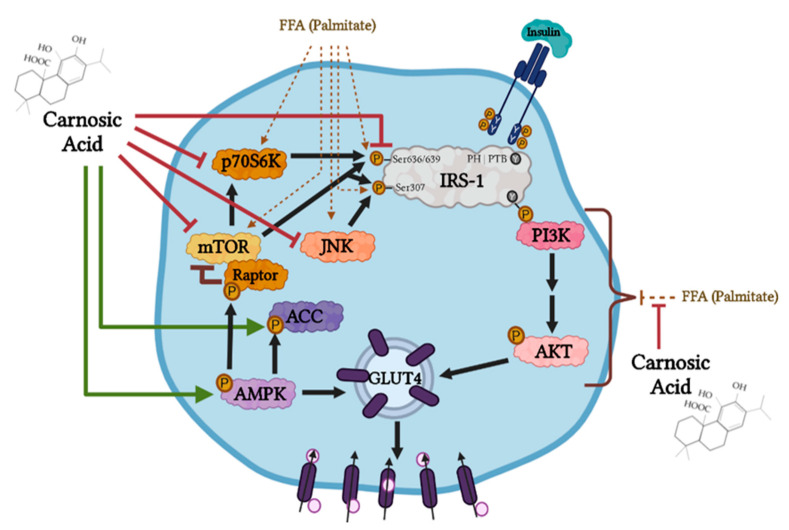
Carnosic acid counteracted the free fatty acid (FFA; palmitate)-induced muscle and fat cell insulin resistance. Carnosic acid prevented the palmitate-induced phosphorylation/activation of JNK, mTOR, and p70S6K, while the activation of AMPK and phosphorylation of ACC was increased. Under elevated free fatty acid conditions, carnosic acid restored the insulin stimulated Akt phosphorylation/activation, GLUT4 plasma membrane translocation, and glucose uptake. Created with BioRender.com. Green and red arrows represent the findings of the present study. Black and brown arrows represent established common knowledge.

## Data Availability

Data supporting reported results can be provided upon request.

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
