# Peer review of "Carnosic Acid Attenuates the Free Fatty Acid-Induced Insulin Resistance in Muscle Cells and Adipocytes"

_cells, 2022, doi:10.3390/cells11010167_

Round 1
Reviewer 1 Report
The authors presented a paper describing the effects of carnosic acid (CA), a natural compound present in rosemary, on the IR induced by palmitate in L6 myotubes and in 3T3L1 differentiated murine adipocytes.
The data indicate that CA is able to attenuate IR induced by palmitate, as suggested by the analysis of the levels of AKT phosphorylation induced by insulin and of the GLUT4 translocation on the plasmatic membrane.
To explain the mechanism of action, the authors analyzed the effects of CA on the IRS1 phosphorylation levels. In addition, the authors investigated the effects of CA on the activation level of mTOR, P70S6K, JNK, known to stimulate IR by phosphorylating IRS1, and of AMPK, whose role could be to inhibit mTOR or P70S6K.
Major concerns:
-With some exceptions, the data are convincing on the experimental results, even if they are almost descriptive. The conclusions require specific control experiments that must be carried out in the context of cellular biology/physiology. For example, there are no data indicating that the activation of AMPK by CA causes a reduction in the levels of P-IRS1 increased following the treatment of FFA. This requires the dedicated experiments with an AMPK inhibitor or, with gene silencing. The same goes for JNK, mTOR and P70S6K as well.
-Most of the studies are conducted in L6 myotubes, only one experiment was conducted in 3T3L1. It would therefore be necessary to repeat the experiments in 3T3L1, as clearly indicated by the same title of the manuscript.
-It is unclear why the authors performed the immunoprecipitation experiments to determine the phosphorylation levels of IRS1.
In Figure 7A, the levels of P-AMPK are clearly higher in the palmitate-treated sample than in the control, while the authors state that the levels of P-AMPK remain unchanged. A more convincing blot should be provided.
In conclusion, the data presented are descriptive and further control experiments are necessary to prove with certainty the conclusions reached by the authors. This is crucial for a cell biology journal like “Cells”.
Minor
- Figure 7A incorrectly indicates P-ACC-Thr172
As indicated by the author instructions:”In order to ensure the integrity and scientific validity of blots (including but not limited to Western blots) and gel data reporting, original, uncropped and unadjusted images should be uploaded as Supporting Information files at time of initial submission”
Author Response
Reviewer 1:
The authors presented a paper describing the effects of carnosic acid (CA), a natural compound present in rosemary, on the IR induced by palmitate in L6 myotubes and in 3T3L1 differentiated murine adipocytes.
The data indicate that CA is able to attenuate IR induced by palmitate, as suggested by the analysis of the levels of AKT phosphorylation induced by insulin and of the GLUT4 translocation on the plasmatic membrane.
To explain the mechanism of action, the authors analyzed the effects of CA on the IRS1 phosphorylation levels. In addition, the authors investigated the effects of CA on the activation level of mTOR, P70S6K, JNK, known to stimulate IR by phosphorylating IRS1, and of AMPK, whose role could be to inhibit mTOR or P70S6K.
Major concerns:
-With some exceptions, the data are convincing on the experimental results, even if they are almost descriptive. The conclusions require specific control experiments that must be carried out in the context of cellular biology/physiology. For example, there are no data indicating that the activation of AMPK by CA causes a reduction in the levels of P-IRS1 increased following the treatment of FFA. This requires the dedicated experiments with an AMPK inhibitor or, with gene silencing. The same goes for JNK, mTOR and P70S6K as well.
In an attempted to elucidate the mechanistic link between AMPK activation and mTOR inhibition, we used specific antibodies and examined raptor, a downstream target of AMPK and regulator of mTOR activity.
These new data are included in the text and Figure 6C of the revised manuscript (see below)
The activity of mTOR is influenced by raptor (regulatory-associated protein of mammalian target of rapamycin (mTOR). Phosphorylation of raptor on S792 inhibits mTOR [70,71]. We examined both phosphorylated (S792) and total raptor. Treatment with CA alone (CA: 169 % of control) or in the presence of palmitate (CA+P: 239% of P) increased raptor phosphorylation (Figure 6C).
The following is added to the discussion
To elucidate the mechanistic link between AMPK activation and mTOR inhibition by CA we examined raptor. Raptor is a constitutively binding protein of mTOR complex 1 (mTORC1) and plays a role in regulating mTOR activity. Importantly, activated AMPK directly phosphorylates raptor on Ser722/792 [70,71], leading to inhibition of mTOR activity. Treatment with CA increased AMPK phosphorylation/activation and increased raptor phosphorylation suggesting that the inhibition of mTOR phosphorylation /activation is mediated by AMPK activation.
-Most of the studies are conducted in L6 myotubes, only one experiment was conducted in 3T3L1. It would therefore be necessary to repeat the experiments in 3T3L1, as clearly indicated by the same title of the manuscript.
We followed the reviewer’s suggestion and performed additional experiments utilizing 3T3-L1 adipocytes. The new data (seen below) are included in section 3.8 and Figures 8, 9 and 10 of the revised manuscript. (Please note that 3T3L1 adipocytes require at least 10-12 days to differentiate and this significantly affects the time required for completion of the revisions).
3.8. Carnosic Acid prevents the effects of Palmitate in 3T3-L1 Adipocytes
Similarly to the experiments we conducted in L6 myotubes, we also examined the effect of CA on signaling molecules in palmitate-treated 3T3-L1 adipocytes. Treatment of 3T3-L1 adipocytes with insulin resulted in a significant increase in Akt (Ser473 residue) phosphorylation (I: 225.5 ± 33.16% of control, p = 0.0387, Figure 8A) and this response was abolished in the presence of palmitate (P+I: 73.35 ± 43.77% of control, p = 0.0155, Figure 8A). However, in the presence of CA, the decline in the insulin-stimulated Akt phosphorylation seen with palmitate was completely prevented (CA+P+I: 272.8 ± 42.01% of control, p < 0.01, Figure 8A). Palmitate alone did not have a significant effect on the basal Akt phosphorylation. The total levels of Akt were not significantly affected by any of the treatments (I: 108.6 ± 11.19%, P: 114 ± 11.66%, P+I: 102.7 ± 11.75%, CA+P+I: 103.9 ± 13.31% of control, Figure 8A).
Exposure of the cells to palmitate (0.4 mM, 16 h) significantly increased serine phosphorylation of IRS-1 (P:151 ± 18.89% of control, p < 0.01, Figure 8B) and treatment with CA abolished this palmitate-induced effect (CA+P: 54.75 ± 9.56% of control, p < 0.01, Figure 8B). CA alone did not affect phosphorylation of IRS-1, and the total levels of IRS-1 were not significantly changed by any treatment.
(A)
(B)
Figure 8. Effects of palmitate and carnosic acid on Akt expression and phosphorylation/activation and serine phosphorylation and expression of IRS-l in adipocytes. Fully differentiated 3T3-L1 adipocytes were treated without (control, C) or with 0.4 mM palmitate (P) for 16 h in the absence or the presence of 20 μM carnosic acid (CA) followed by stimulation without or with 100 nM insulin (I) for 30 min. After treatment, the cells were lysed, and SDS-PAGE was performed, followed by immunoblotting with specific antibodies that recognize phosphorylated Ser473 or total Akt (A) and Ser306 or total IRS-1 (B). Representative immunoblots are shown. The densitometry of the bands was measured and expressed in arbitrary units. The data are the mean ± SE of four separate experiments (* p < 0.05, ** p < 0.01 vs. control, # p < 0.05, ## p < 0.01 as indicated).
Exposure of the cells to palmitate significantly increased JNK (P:205.2 ± 21.34 % of control, p < 0.05, Figure 9A) mTOR (P: 209.2 ± 20.5.% of control, p = 0.002 Figure 9B) and p70S6K (P: 259.4 ± 24.33% of control, p < 0.0001, Figure 9C) phosphorylation. Treatment with CA completely abolished the palmitate-induced phosphorylation of JNK (CA+P: 87.25 ± 29.20 % of control, p < 0.05, Figure 9A), mTOR (CA+P: 97.41 ± 26.22% of control, p = 0.0016, Figure 9B) and p70S6K (CA+P: 98.96 ± 3.252% of control, p < 0.0001, Figure 9C). Treatment with CA alone did not have an effect on the basal JNK (CA: 110.1 ± 12.78 % of control, p < 0.0001, Figure 9A), mTOR (CA: 110.7 ± 9.618% of control, p = 0.97, Figure 9B) or p70S6K (CA = 124.8 ± 5.623 % of control, p < 0.0001, Figure 9C) phosphorylation. The total levels of JNK (Figure 9A), mTOR (Figure 9B), and p70S6K (Figure 9C), were not significantly changed by any treatment.
(A)
(B)
(C)
Figure 9. Effects of palmitate and carnosic acid on JNK, mTOR, and p70S6K expression and phosphorylation in adipocytes. Fully differentiated adipocytes were treated without (control, C) or with 0.4 mM palmitate (P) for 16 h in the absence or the presence of 20 μM carnosic acid (CA). After treatment, the cells were lysed, and SDS-PAGE was performed, followed by immunoblotting with specific antibodies that recognize phosphorylated Thr183/Tyr185 or total JNK (A), phosphorylated Ser2448 or total mTOR (B) and phosphorylated Thr389 or total p70S6K (C). Representative immunoblots are shown. The densitometry of the bands was measured and expressed in arbitrary units. The data are the mean ± SE of four separate experiments (* p < 0.05, ** p < 0.01, *** p < 0.001 vs. control, # p < 0.05, ## p < 0.01, ### p < 0.001 vs. palmitate alone).
We also investigated the effect of CA on adipocyte AMPK and ACC. We performed a dose-response experiment, as to our knowledge, such experiments with CA were not previously performed in adipocytes. Treatment of 3T3-L1 adipocytes with 2, 10 or 20 μM CA for 16 h resulted in a dose-dependent increase in the phosphorylation/activation of AMPK(CA 2 μM: 140.3 ± 12.77%, CA 10 μM: 263.2 ± 41.12%, CA 20 μM: 410.2 ± 97.98%, Figure 10A) and ACC (CA 2 μM: 141.3 ± 11.42%, CA 10 μM: 229.8 ± 13.58%, CA 20 μM: 240 ± 33.64% of control, Figure 10B). Importantly, CA increased the phosphorylation of AMPK (CA 20 μM +P: 425.9 ± 141.6% of control, p < 0.05) and ACC (CA+P: 253.5 ± 42.07% of control, p < 0.01), even in the presence of palmitate (Figure 10A, B). Treatment with palmitate alone had no significant effect on the phosphorylation of AMPK or ACC (P: 143 ± 28.91% and 128.2 ± 35.05% of control, respectively, Figure 10A, B). Furthermore, the total levels of AMPK or ACC were not affected by any treatment (P: 85.70 ± 13.17% and 107.6 ± 37.61%, CA: 79.05 ± 12.58% and 133.8 ± 33.31%, CA+P: % and 102.8 ± 17.29% of control, respectively, Figure 10 A, B).
(A)
(B)
Figure 10. Effects of palmitate and carnosic acid on AMPK expression and phosphorylation in adipocytes. Fully differentiated adipocytes were treated without (control, C) or with 0.4 mM palmitate (P) for 16 h in the absence or the presence of 20 μM carnosic acid (CA). After treatment, the cells were lysed, and SDS-PAGE was performed, followed by immunoblotting with specific antibodies that recognize phosphorylated Thr172 or total AMPK and Ser79 or total ACC. Representative immunoblots are shown (A). The densitometry of the bands was measured and expressed in arbitrary units (B). The data is the mean ± SE of three separate experiments. (* p < 0.05, ** p < 0.01, *** p < 0.001 vs. control, # p < 0.05, ## p < 0.01 vs. palmitate alone).
-It is unclear why the authors performed the immunoprecipitation experiments to determine the phosphorylation levels of IRS1.
We had difficultly detecting IRS-1 (very faint bands were detected) in total L6 cell lysates and that is why we took the approach to immunoprecipitate it. In the past others [1,2] haven taken a similar approach to immunoprecipitated IRS-1. For the 3T3L1 experiments we used whole cell lysates.
- Frendo-Cumbo, S.; Jaldin-Fincati, J.R.; Coyaud, E.; Laurent, E.M.N.; Townsend, L.K.; Tan, J.M.J.; Xavier, R.J.; Pillon, N.J.; Raught, B.; Wright, D.C.; et al. Deficiency of the Autophagy Gene ATG16L1 Induces Insulin Resistance through KLHL9/KLHL13/CUL3-Mediated IRS1 Degradation. J. Biol. Chem. 2019, 294, 16172–16185, doi:10.1074/jbc.RA119.009110.
- Huang, C.; Thirone, A.C.P.; Huang, X.; Klip, A. Differential Contribution of Insulin Receptor Substrates 1 versus 2 to Insulin Signaling and Glucose Uptake in L6 Myotubes. J. Biol. Chem. 2005, 280, 19426–19435, doi:10.1074/jbc.M412317200.
In Figure 7A, the levels of P-AMPK are clearly higher in the palmitate-treated sample than in the control, while the authors state that the levels of P-AMPK remain unchanged. A more convincing blot should be provided.
Thank You. We have provided a better blot. Please see figure 7 of the revised manuscript. (added below)
(A)
(B)
Figure 7. Effects of palmitate and carnosic acid on AMPK and ACC expression and phosphorylation. Fully differentiated myotubes were treated without (control, C) or with 0.2 mM palmitate (P) for 16 h in the absence or the presence of 2 μM carnosic acid (CA). After treatment, the cells were lysed, and SDS-PAGE was performed, followed by immunoblotting with specific antibodies that recognize phosphorylated Thr172,total AMPK, phosphorylated Ser79 or total ACC. Representative immunoblots are shown (A). The densitometry of the bands was measured and expressed in arbitrary units (B). The data is the mean ± SE of three separate experiments. (** p < 0.01, *** p < 0.001 vs. control, # p < 0.05 vs. palmitate alone).
In conclusion, the data presented are descriptive and further control experiments are necessary to prove with certainty the conclusions reached by the authors. This is crucial for a cell biology journal like “Cells”.
The revised manuscript includes additional experiments/data examining the effects on raptor. The revised manuscript includes additional extensive experiments utilizing adipocytes.
Minor
- Figure 7A incorrectly indicates P-ACC-Thr172
Thank You. This has been corrected

Reviewer 2 Report
In obese people, increased plasma lipid levels lead to insulin resistance, which ultimately results in type 2 diabetes mellitus (T2DM). In this manuscript, Hartogh et al analysed the effect of carnosic acid (CA), which is found in rosemary extract, on the free fatty acid (FFA)-induced insulin resistance in muscle cells (L6 myotubes) and adipocytes (3T3 L1 adipocytes). The authors found that CA restores insulin-stimulated glucose-uptake, GLUT4 translocation and PKB phosphorylation in palmitate treated muscle cells and 3T3 L1 adipocytes. In addition, they also showed that CA prevents palmitate-induced phosphorylation of IRS-1, JNK, mTOR and p70S6K in L6 myotubes. Further, CA increased the phosphorylation of AMPK and ACC in the presence or absence of palmitate. They concluded from these findings that CA can counteract palmitate-induced insulin resistance in muscle cells and adipocytes. Although the data are in agreement with the conclusions, addressing the following comments will strengthen the manuscript.
- In Figures 1-3, the effect of carnosic acid (CA) on glucose uptake in cells stimulated without or with insulin should be included.
- In Figure 7, p-AMPK and total AMPK blots need to be aligned properly.
- There are several typos, particularly in the methods section. For example, glutamate concentration should be 2mM (line 129 on page 3), IBMX concentration should be 0.5mM (line 131 on page 3). In lane 148 on page 4, “containing 10% goat serum containing-PBS” should be changed to “(10% goat serum containing-PBS)”. Also, anti-Myc antibody details are missing in the 2.1 materials section on page 3.
Author Response
Reviewer 2
Comments and Suggestions for Authors
In obese people, increased plasma lipid levels lead to insulin resistance, which ultimately results in type 2 diabetes mellitus (T2DM). In this manuscript, Hartogh et al analysed the effect of carnosic acid (CA), which is found in rosemary extract, on the free fatty acid (FFA)-induced insulin resistance in muscle cells (L6 myotubes) and adipocytes (3T3 L1 adipocytes). The authors found that CA restores insulin-stimulated glucose-uptake, GLUT4 translocation and PKB phosphorylation in palmitate treated muscle cells and 3T3 L1 adipocytes. In addition, they also showed that CA prevents palmitate-induced phosphorylation of IRS-1, JNK, mTOR and p70S6K in L6 myotubes. Further, CA increased the phosphorylation of AMPK and ACC in the presence or absence of palmitate. They concluded from these findings that CA can counteract palmitate-induced insulin resistance in muscle cells and adipocytes. Although the data are in agreement with the conclusions, addressing the following comments will strengthen the manuscript.
- In Figures 1–3, the effect of carnosic acid (CA) on glucose uptake in cells stimulated without or with insulin should be included.
- We thank the reviewer for this comment. We followed the reviewer’s suggestion and performed additional experiments. The new data depicting the effects of CA on basal and insulin-stimulated glucose uptake in both L6 and 3T3-L1 cells are included in the revised manuscript (Figure 1 inserts).
Text added to results section
Line 216-218 for L6 muscle cells
CA treatment alone significantly increased the basal (CA: 253 ± 19.8% of control, p < 0.01 vs. control) and the insulin-stimulated glucose uptake (CA+I: 269 ± 13.2% of control, p < 0.001 vs. control, p < 0.05 vs. insulin, Figure 1A insert).
Line 223-225 for 3T3L1 adipocytes
Treatment with CA significantly increased the basal (CA: 152 ± 4.5% of control, p < 0.001) but not the insulin-stimulated glucose uptake (CA+I: 166 ± 2.4% of control, p < 0.001, Figure 1B insert).
In addition, we performed additional experiments and examined the effect of CA on GLUT4 plasma membrane levels in L6 cells (Figure 2 insert).
Text added to results section line 248-250
Treatment with CA alone did not affect GLUT4 plasma membrane levels (CA: 109.4 ± 3.4% of control, p < 0.05, Figure 2) as seen previously [54] and did not increase the insulin-stimulated GLUT4 translocation (CA+I: 196.2 ± 7.9% of control, Figure 2 insert).
Text added to discussion section line 498-501
CA alone increased glucose uptake in L6 muscle cells (Figure 1A insert) without affecting GLUT4 plasma membrane levels (Figure 2 insert), in agreement with previous finding by our group [54], indicating that at the basal level, CA may influence the intrinsic activity of plasma membrane glucose transporters.
3. In Figure 7, p-AMPK and total AMPK blots need to be aligned properly. We thank the reviewer for this comment. We have changed the representative figure for AMPK and is more aligned now.
4. There are several typos, particularly in the methods section. For example, glutamate concentration should be 2mM (line 129 on page 3), IBMX concentration should be 0.5mM (line 131 on page 3). In lane 148 on page 4, “containing 10% goat serum containing-PBS” should be changed to “(10% goat serum containing-PBS)”. Also, anti-Myc antibody details are missing in the 2.1 materials section on page 3.
We thank the review for this comment and have gone through the methodology section and removed the typos and included the anti-myc antibody details in the materials section.

Round 2
Reviewer 1 Report
The authors presented a new version of the manuscript in which they added new Western blotting experiments in order to answer the main concern indicated in the first revision process, namely the lack of an appropriate control that certifies the validity of the hypothesis (the role of AMPK in mediating the effects of the carnosic acid). In attempting to provide a control experiment, the authors analyzed Raptor phosphorylation, which would act as a negative regulator of mTOR.
-A classic study on transduction pathways requires the use of appropriate controls that demonstrate with certainty the involvement of specific kinases in mediating cellular responses to a stimulus. In the previous review, it was suggested to add a control experiment that confirms the validity of the authors' hypothesis, for example, an inhibitor compound. In this case, an inhibitor of AMPK.
In my opinion, the analysis of Raptor phosphorylation does not represent a valid control, perhaps it represents further observational data that itself requires an adequate control.
-The images of the blots reported as supplementary data are cropped, and do not reflect the requirements of the journal, which require the original, not cropped, images . The authors must provide the full images, with the marker that can be acquired with the Chemidoc (which has been used by the authors).
Reviewer 2 Report
My concerns have been satisfactorily addressed in the revised manuscript.